# Genotype–phenotype analysis of hearing function in patients with DFNB1A caused by the c.-23+1G>A splice site variant of the *GJB2* gene (Cx26)

Fedor M. Teryutin[1], Vera G. Pshennikova[1], Aisen V. Solovyev[2], Georgii P. Romanov[2], Sardana A. Fedorova[1,2], Nikolay A. Barashkov[1] *

**1** Laboratory of Molecular Genetics, Yakut Scientific Centre of Complex Medical Problems, Yakutsk, Russia,
**2** Laboratory of Molecular Biology, Institute of Natural Sciences, M.K. Ammosov North-Eastern Federal University, Yakutsk, Russia

* barashkov2004@mail.ru

**Data Availability Statement:** All relevant data are within the manuscript.

## Abstract

The audiological features of hearing loss (HL) in patients with autosomal recessive deafness type 1A (DFNB1A) caused by splice site variants of the *GJB2* gene are less studied than those of patients with other variants of this gene. In this study, we present the audiological features of DFNB1A in a large cohort of 134 patients with the homozygous splice site variant c.-23+1G>A and 34 patients with other biallelic *GJB2* genotypes (n = 168 patients with DFNB1A). We found that the preservation of hearing thresholds in the speech frequency range ($PTA_{0.5,1.0,2.0,4.0\ kHz}$) in patients with the c.[-23+1G>A];[-23+1G>A] genotype is significantly better than in patients with the "severe" c.[35delG];[35delG] genotype (*p* = 0.005) and significantly worse than in patients with the "mild" c.[109G>A];[109G>A] genotype (*p* = 0.041). This finding indicates a "medium" pathological effect of this splice site variant on hearing function. A detailed clinical and audiological analysis showed that in patients with the c.[-23+1G>A];[-23+1G>A] genotype, HL is characterized as congenital or early onset (57.5% onset before 12 months), sensorineural (97.8%), bilateral, symmetrical (82.8%), variable in severity (from mild to profound HL, median hearing threshold in $PTA_{0.5,1.0,2.0,4.0\ kHz}$ is 86.73±21.98 dB), with an extremely "flat" audioprofile, and with a tendency toward slow progression (a positive correlation of hearing thresholds with age, *r* = 0.144, *p* = 0.041). In addition, we found that the hearing thresholds in $PTA_{0.5,1.0,2.0,4.0\ kHz}$ were significantly better preserved in females (82.34 dB) than in males (90.62 dB) (*p* = 0.001). We can conclude that in patients with DFNB1A caused by the c.-23+1G>A variant, male sex is associated with deteriorating auditory function; in contrast, female sex is a protective factor.

## Introduction

Currently, one in 500–1000 children is born with significant hearing loss (HL), and up to 50% of these cases have a hereditary etiology [1–3]. However, the genetic causes of most congenital

**Funding:** This study was supported by the Ministry of Science and Higher Education of the Russian Federation (FSRG-2023-0003) (AVS., GPR and SAF) and the YSC CMP project "Study of the genetic structure and burden of hereditary pathology of the populations of the Republic of Sakha (Yakutia)" (to FMT, VGP, and NAB). There was no additional external funding received for this study.

**Competing interests:** The authors have declared that no competing interests exist.

cases of HL are extremely heterogeneous [3]. Thus, the proportion of pathogenic biallelic variants of the *GJB2* gene (13q11-q12) [4] encoding the intercellular gap junction protein connexin 26 (Cx26) [5, 6] leading to autosomal recessive deafness type 1A (DFNB1A, OMIM #220290) [7] is significant, and in some populations, it is up to 50–60% [8–10]. More than 380 pathogenic variants have been found in the *GJB2* gene [Hereditary HL Homepage: http://hereditaryhearingloss.org/, accessed 20.12.2023].

In the past, researchers generally believed that the pathological mechanisms underlying DFNB1A were caused by biallelic pathogenic variants in the *GJB2* gene, which damage the homomeric (consisting of Cx26) or heteromeric (consisting of different connexin proteins) gap junction channels [11]. This was thought to lead to a $K^+$ circulation defect and abnormal ATP-$Ca^{2+}$ signals in the cochlea [6]. It has been hypothesized that gap junctions in the cochlea, especially those involving Cx26, provide an intercellular passage by which $K^+$ is transported to maintain high levels of the endocochlear potential, which is essential for sensory hair cell excitation [6]. However, subsequent studies have shown that $K^+$ circulation is rarely associated with the pathological process of DFNB1A [12–16]. Researchers now believe that pathogenic Cx26 variants cause changes in $Ca^{2+}$ signaling and ATP release, as well as columnar cell cytoskeletal developmental disorders, all of which contribute to the occurrence of HL [12–17]. However, the exact pathological mechanism of DFNB1A remains unknown [17].

Moreover, a large systematic analysis of 270 unrelated patients with biallelic *GJB2* pathogenic variants (30 Belgian, 131 Italian, 42 Spanish, and 74 American patients) [18] and a multi-center study of 1,531 patients with biallelic *GJB2* pathogenic variants (90% of participants were of Caucasian origin) [19] as well as a meta-analysis of more than 200 original articles [8] on the genotype–phenotype features of hearing function in patients with DFNB1A showed extremely variable hearing phenotypes that ranged from mild to profound HL [8, 18, 19]. However, it has been found that HL is significantly more severe in patients with biallelic truncating variants (T/T—leading to premature stop codons and disruption of splice sites) than in compared to patients with nontruncating biallelic variants (NT/NT—leading to amino acid substitutions) [19]. Currently, among the truncating variants, the most thoroughly studied are single nucleotide deletions, c.35delG p.(Gly12ValfsX2) and c.235delC p.(Leu79Cysfs*3) [8, 18–46]. The genotype–phenotype features of homozygous patients are well described for nontruncating hypomorphic c.101T>C p.(Met34Thr) and c.109G>A p.(Val37Ile) missense variants with low penetrance and weak pathogenic potential [25, 36, 40, 45, 47–60]. However, the genotype–phenotypic features of some truncating variants of this gene affecting splicing sites are less studied. According to ClinVar (https://www.ncbi.nlm.nih.gov/clinvar/?term=gjb2%5Bgene%5D&redir=gene, accessed on 20.12.2023), only two (pathogenic or likely pathogenic) splicing site variants are known in the *GJB2* gene (donor splice site—c.-23+1G>A [61, 62] and acceptor splice site variant c.-22-2A>C) [63]. Studies of the genotype–phenotype correlations of different molecular effects of *GJB2* variants are important for understanding the clinical features of different allelic forms of DFNB1A and may help to reveal the pathological process of this disease, which is crucial for developing targeted treatments.

An extremely high prevalence of the c.-23+1G>A variant, due to a founder effect, was identified among the Turkic-speaking Yakut population living in the Sakha Republic, which is located in the Siberian part of Russia [64, 65]. The prevalence of DFNB1A caused by the c.-23+1G>A variant of the *GJB2* gene was 16.2 per 100,000 people in this region of Russia, while carrier frequency varied from 3% to 11% among different indigenous populations of Eastern Siberia [64]. In a recent study on spectrum and frequency of pathogenic variants of the DFNB1 locus in a large Russian cohort of patients with nonsyndromic sensorineural HL (2,569 unrelated individuals), 39 pathogenic and likely pathogenic variants of the *GJB2* gene were identified [66]. Among them, the c.-23+1G>A splice site variant was third most frequent

across Russia (4.4%) [66]. Despite relatively high prevalence of this variant in Russia, there are still no detailed studies on the genotype-phenotype characteristics of individuals with the c.-23+1G>A variant of the *GJB2* gene. Currently, only one retrospective report of the audiological characteristics of 40 patients with the c.[-23+1G>A];[-23+1G>A] genotype is known [64].

In this study, we present a detailed audiological analysis of a large cohort of 134 DFNB1A patients with the homozygous donor splice site variant c.-23+1G>A in the *GJB2* gene and 34 patients with other biallelic *GJB2* genotypes (n = 168 patients with DFNB1A).

## Materials and methods

### Study sample

The sample of HL individuals consisted of patients from the Republican Hospital #1 of the National Medical Centre (Yakutsk, Russian Federation), students at the Republican special residential schools for deaf and hard-of-hearing children (Yakutsk, Russian Federation) and members of the Yakutsk Department of the All-Russian Society of the Deaf (Yakutsk, Russia), as previously described [65, 67]. In total, complete sequencing of the noncoding and coding regions of the *GJB2* gene was performed for 514 patients with different degrees of HL. A total of 168 DFNB1A patients with biallelic *GJB2* variants and no objective otological problems were included in this study. The study sample was represented by the following *GJB2* genotypes: c.[-23+1G>A];[-23+1G>A] (n = 134), c.[35delG];[35delG] (n = 11), c.[109G>A]; [109G>A] (n = 2), c.[-23+1G>A];[109G>A] (n = 2), c.[-23+1G>A];[35delG] (n = 14), c. [35delG];[313_326del14] (n = 1), c.[-23+1G>A];[167delT] (n = 1), c.[-23+1G>A]; [313_326del14] (n = 1), c.[-23+1G>A];[333_334delAA] (n = 1), and the c.[35delG];[del (GJB6-D13S1830)] (n = 1). The largest group of patients in our cohort had the c.[-23+1G>A]; [-23+1G>A] genotype (134 patients), so it was used as a reference (Ref). For audiological analysis, we used the number of ears (168 patients, 336 ears).

### Hearing status

All patients underwent medical examinations, including a collection of complaints and a medical history. Hearing status was confirmed by an audiological study, including tuning fork tests (tuning fork C128 Hz, KaWe, Asperg, Germany), impedance audiometry and threshold tone audiometry (tympanometer and audiometer AA222, Interacoustics, Middelfart, Denmark) using air conduction at frequencies of 0.25, 0.5, 1.0, 2.0, 4.0, and 8.0 kHz and by bone conduction at frequencies of 0.25, 0.5, 1.0, and 4.0 kHz with steps of 5.0 dB. For detailed audiological analysis, we used the clinically important speech frequency range in pure tone averages ($PTA_{0.5,1.0,2.0,4.0 \text{ kHz}}$). Five children, due to their young age, were tested by the ASSR test (Audera, Grason-Stadler, Eden Prairie, MN, USA). Audiograms that had breaks were normalized by introducing the maximum readings (120.0 dB) at frequencies where the patient did not respond. The type of HL was sensorineural with an increase in bone and air conduction thresholds on audiograms, mixed with an increase in bone and air conduction thresholds with an interval exceeding a total of 20.0 dB in $PTA_{0.5,1.0,2.0,4.0 \text{ kHz}}$. HL was considered asymmetric when the interaural difference in hearing thresholds at $PTA_{0.5,1.0,2.0,4.0 \text{ kHz}}$ was greater than 15.0 dB. The degree of HL was assessed by the average hearing threshold in $PTA_{0.5,1.0,2.0,4.0 \text{ kHz}}$ according to the classification by Clark, J. G. (1981) [68]: normal–from 10 to 15 dB, slight– from 16 to 25 dB, mild–from 26 to 40 dB (I degree), moderate–from 41 to 55 dB (II degree), moderately severe–from 56 to 70 dB (III degree), severe–from 71 to 90 dB (IV degree), and profound > 90 dB (deafness).

## Detection of *GJB2* genotypes

DNA was extracted from blood leukocytes via the phenol–chloroform method. Amplification of the coding (exon 2), noncoding (exon 1) and flanking intronic regions of the *GJB2* gene was performed by PCR on a T100 thermocycler (Bio-Rad, Hercules, NY, USA) using the following primers: 5′-CCGGGAAGCTCTGAGGAC-3′ and 5′-GCAACCGCTCTGGGTCTC-3′ for amplification of exon 1 [69]; and 5′-TCGGCCCCAGTGGTACAG-3′ and 5′-CTGGGCAATGCGTTA AACTGG-3′ for amplification of exon 2 [7, 70, 71]. The PCR products were subjected to Sanger sequencing using the same primers on an ABI PRISM 3130XL (Applied Biosystems, Waltham, MA, USA) at the Genomics Core Facility of Institute of Chemical Biology and Fundamental Medicine, Siberian Branch of the Russian Academy of Sciences (Novosibirsk, Russia). DNA sequence variations were identified by comparison with the *GJB2* gene reference sequences chr13 (GRCh38.p13), NC_000013.11, NG_008358.1, NM_004004.6 and NP_003995.2 (NCBI, Gene ID: 2706).

The large DFNB1 deletions were screened using oligonucleotide primers for the detection of a breakpoint junction fragment specific for 309 kb-del(*GJB6*-D13S1830)—*GJB6* F5′-TTTAGGGCATGATTGGGGTGATTT-3′ and R5′-CACCATGCGTAGCCTTAACCATTT-3′ [72]; and for 232 kb-del(*GJB6*-D13S1854) F5′-TCATAGTGAAGAACTCGATGCTGTTT-3′ and R5′-CAGCGGCTACCCTAGTTGTGGTT-3′ [72]; with an internal control fragment (*GJB6*, exon 1) F5'-CGTCTTTGGGGGTGTTGCTT-3' and R5'-CATGAAGAGGGCGTACAAGTTAGAA-3' (*GJB6*, exon 1). Screening of the 101 kb—del(*GJB2*-d13S175) region was performed using oligonucleotide primers for the detection of the breakpoint junction fragment F5′-GCTCTGCCCAGAT GAAGATCTC-3′ and R5′-CCTTCCAGGAGAGTTCACAACTC-3′ with the internal control fragment F5′-GTGATTCCTGTGTTGTGTGCATTC-3′ and R5′-CCTCATCCCTCTCATGCTGTC-3′ (*GJB2*, exon 2) [66].

## Statistical analysis

Statistical analysis of the clinical and audiological data in patients with the c.[-23+1G>A];[-23+1G>A] genotype was performed using the Sampling program, kindly provided by M. Macaulay and adapted by M. Metspalu. Differences of the credible interval at the 95% significance level were considered statistically significant. Comparison of hearing thresholds in $PTA_{0.5,1.0,2.0,4.0\ kHz}$ of the reference group c.[-23+1G>A];[-23+1G>A] with other *GJB2* genotypes was performed with a Mann–Whitney U test using by software STATISTICA version 8.0 (StatSoft Inc, USA). Differences were considered statistically significant at $p<0.05$. Correlation analysis of hearing thresholds in $PTA_{0.5,1.0,2.0,4.0\ kHz}$ with age in patients with c.[-23+1G>A];[-23+1G>A] genotype was performed with a r-linear regression analysis using by STATISTICA version 8.0 (StatSoft Inc, USA). Differences were considered statistically significant at $p<0.05$. Statistical analysis of the hearing thresholds between male and female patients with c.[-23+1G>A];[-23+1G>A] genotype was performed with a Student's t-test using by software STATISTICA version 8.0 (StatSoft Inc, USA). Differences were considered statistically significant at $p<0.05$.

## Ethical approval

Written informed consent was obtained from all patients participating in the study. The study was conducted according to the guidelines of the Declaration of Helsinki and approved by the local Biomedical Ethics Committee at the Yakut Scientific Center of Complex Medical Problems, Yakutsk, Russia (Yakutsk, protocol No. 16 of 16 April 2009).

## Results

### Audioprofiles of the ten *GJB2* genotypes at six measured frequencies

We analyzed the audioprofiles at six measured frequencies among 168 patients with 10 different biallelic variants of the *GJB2* gene. The "flat" audioprofiles were detected in patients with genotypes c.[-23+1G>A];[-23+1G>A] (the slope in $PTA_{0.5,1.0,2.0,4.0}$ kHz was only 5.0 dB) and c.[-23+1G>A];[109G>A] (the slope in $PTA_{0.5,1.0,2.0,4.0}$ kHz was only 10.0 dB) (Fig 1). The "sloping" audioprofile was found in patients with genotypes c.[-23+1G>A];[35delG], c.[-23+1G>A]; [313_326del14], c.[109G>A];[109G>A], c.[35delG];[35delG], c.[35delG];[del(*GJB6*-D13S1830)], and c.[35delG];[313_326del14] (the slope in $PTA_{0.5,1.0,2.0,4.0}$ kHz varied from 12.5 to 20.0 dB) (Fig 1). A "downsloping" audioprofile was found among patients with the c.[-23+1G>A];[167delT] and c.[-23+1G>A];[333_334delAA] genotypes (the slope in $PTA_{0.5,1.0,2.0,4.0}$ kHz varied from 30.0 to 42.5 dB) (Fig 1).

### $PTA_{0.5,1.0,2.0,4.0 \text{ kHz}}$ hearing thresholds in patients with ten different *GJB2* genotypes

To compare the hearing thresholds in $PTA_{0.5,1.0,2.0,4.0 \text{ kHz}}$ of the 10 different *GJB2* genotypes, we used the values of hearing thresholds in individuals with the c.[-23+1G>A];[-23+1G>A] genotype (n = 268 ears) as a reference (Ref). In patients with the c.[35delG];[35delG] genotype, the hearing thresholds at $PTA_{0.5,1.0,2.0,4.0 \text{ kHz}}$ were significantly higher than in the reference group ($p = 0.005$) (Fig 2). In contrast, in patients with the c.[109G>A];[109G>A] ($p = 0.041$) and c.[-23+1G>A];[109G>A] ($p = 0.000$) genotypes, the hearing thresholds at $PTA_{0.5,1.0,2.0,4.0 \text{ kHz}}$ were significantly lower than in the reference group. In patients with the following genotypes: c.[-23+1G>A];[35delG], c.[35delG];[313_326del14], c.[-23+1G>A];[167delT], .[-23+1G>A];[313_326del14], .[-23+1G>A];[333_334delAA], and c.[35delG];[del(*GJB6*-D13S1830)], the hearing thresholds in $PTA_{0.5,1.0,2.0,4.0 \text{ kHz}}$ did not significantly differ from those in the reference group (Fig 2).

Since DFNB1A in patients is caused by different *GJB2* variants, their hearing thresholds differed among the audioprofiles ("flat", "sloping", and "downsloping") (Fig 1), and there was variable preservation of the hearing thresholds in the $PTA_{0.5,1.0,2.0,4.0 \text{ kHz}}$ (Fig 2). To avoid statistical distortions, we aligned our cohort of patients with one biallelic truncating c.[-23+1G>A];[-23+1G>A] genotype (T/T). We excluded individuals with different biallelic truncating and nontruncating (T/T, T/NT and NT/NT) *GJB2* genotypes (n = 34) from further analysis, and detailed audiological analysis was performed only in this most representative cohort of patients with the c.[-23+1G>A];[-23+1G>A] genotype (n = 134 individuals). The audiological data of patients with other biallelic *GJB2* genotypes are presented in the S1 Fig.

### Clinical and audiological analysis in patients with the c.[-23+1G>A];[-23+1G>A] genotype

Among the c.-23+1G>A homozygous patients, 52.9% had a family history of HL, 38.8% had no family history, and the remaining 8.2% had an unknown family history (Fig 3A). HL was detected within 12 months after birth in 57.5% of patients, before 3 years in 5.2%, after 4 years in 7.5%, and in 29.8%, the debut of HL was unknown (Fig 3B). The degree of HL in patients with the c.[-23+1G>A];[-23+1G>A] genotype was mild (2.2%), moderate (10.4%), moderately severe (22.3%), severe (21.6%), and profound (43.2%) (Fig 3C). The sensorineural type of HL was found in 97.8% of patients, and a mixed type of HL was detected in 2.2% of patients (Fig 3D). In 82.8% of the patients, the HL was symmetrical, and in 17.2% of the patients, it was asymmetric (the interaural difference in hearing thresholds exceeded 15.0 dB) (Fig 3E).

## "Flat" audioprofiles

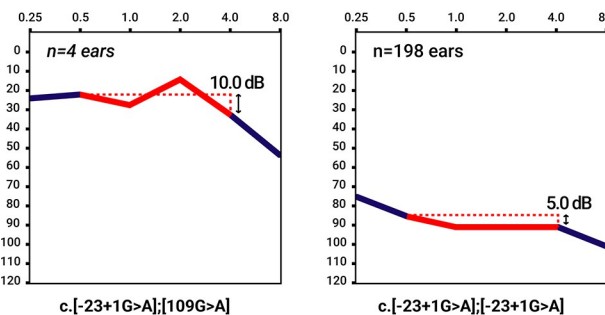

## "Sloping" audioprofiles

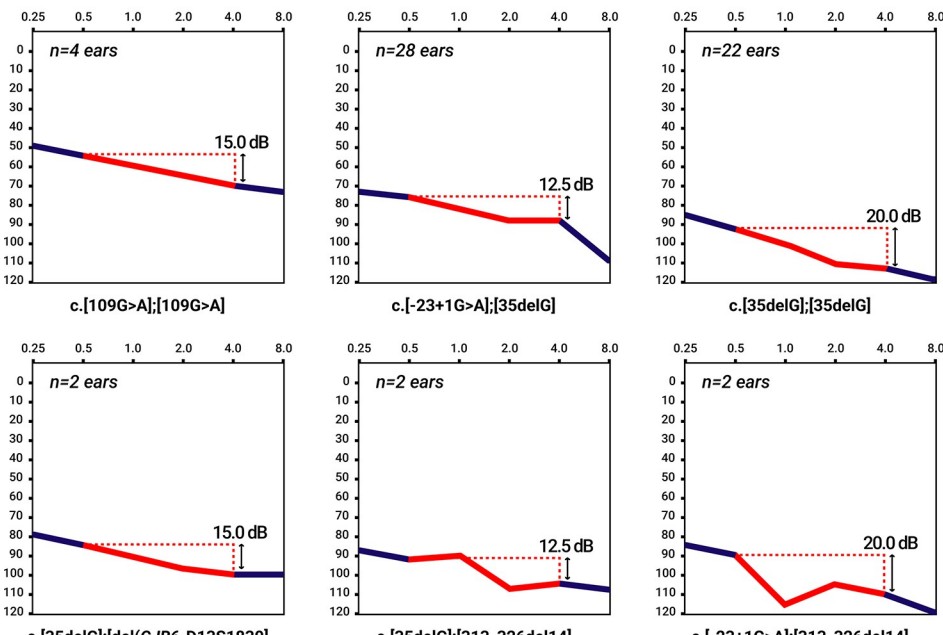

## "Down-sloping" audioprofiles

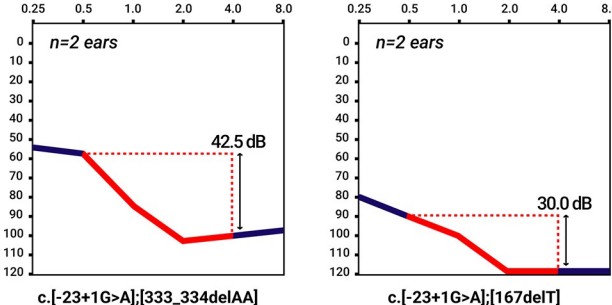

**Fig 1. Audioprofiles of 10 different *GJB2* genotypes at six measured frequencies.**

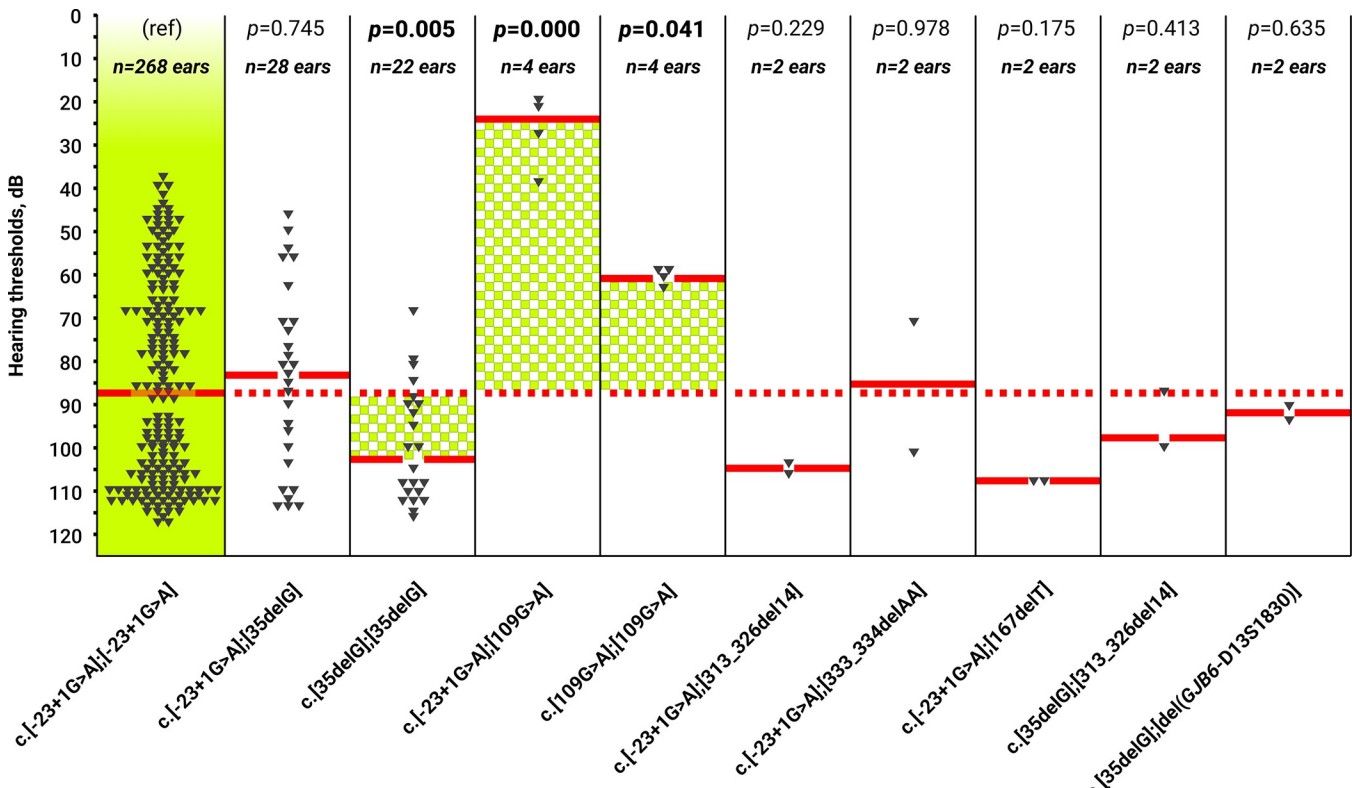

**Fig 2. Comparison of hearing thresholds in PTA$_{0.5,1.0,2.0,4.0 \text{ kHz}}$ of the reference group c.[-23+1G>A];[-23+1G>A] with other *GJB2* genotypes. Note**: the *GJB2* genotypes with statistically significant differences ($p<0.05$) are highlighted by bold font.

A generalized audiogram of patients with the c.[-23+1G>A];[-23+1G>A] genotype (n = 268 ears) demonstrates that the hearing thresholds are variable and have an almost uniform spread of HL from 15.0 dB to 120.0 dB at a low frequency of 0.25 kHz (median of 75.0 dB) and from 45.0 dB to 120.0 dB at a high frequency of 8.0 kHz (median of 120.0 dB). Thus, the median decreased but full levelling off at hearing thresholds in PTA$_{0.5,1.0,2.0,4.0 \text{ kHz}}$ (the slope of the median in PTA$_{0.5,1.0,2.0,4.0 \text{ kHz}}$ was 10.0 dB; the median in the PTA$_{0.5,1.0,2.0,4.0 \text{ kHz}}$ was 88.1 dB with an average hearing threshold of 86.73 dB±21.98 dB) (Fig 4A). The frequency in the group with the c.[-23+1G>A];[-23+1G>A] genotype (n = 268 ears) demonstrated a tendency toward the preservation of low frequencies and greater damage to high frequencies (Fig 4B). It should be noted that at a frequency of 8.0 kHz, sound perception was preserved in half of the studied patients (49.6%) (Fig 4B); therefore, half of the patients in the studied cohort had uniform damage to the cochlea.

## Correlation analysis of hearing thresholds in PTA$_{0.5,1.0,2.0,4.0 \text{ kHz}}$ with age in patients with c.[-23+1G>A];[-23+1G>A] genotype

Correlation analysis of PTA$_{0.5,1.0,2.0,4.0 \text{ kHz}}$ hearing thresholds with age (from 0 to 30 years) was carried out for 99 individuals with the c.[-23+1G>A];[-23+1G>A] genotype (Fig 5). The results of this analysis revealed a linear regression of hearing thresholds in PTA$_{0.5,1.0,2.0,4.0 \text{ kHz}}$ with age, and hearing acuity decreased with older age ($r = 0.144$, $p = 0.041$) (Fig 5A). This sample was also stratified by sex. After stratification by sex, the regression of hearing thresholds was not statistically confirmed, but the general trend of increasing hearing thresholds remained (Fig 5B).

**A  Family history**

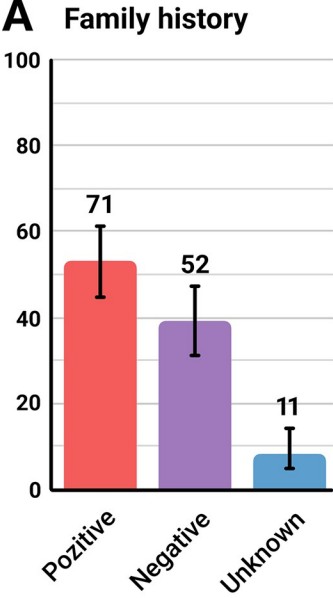

**B  Debut of HL**

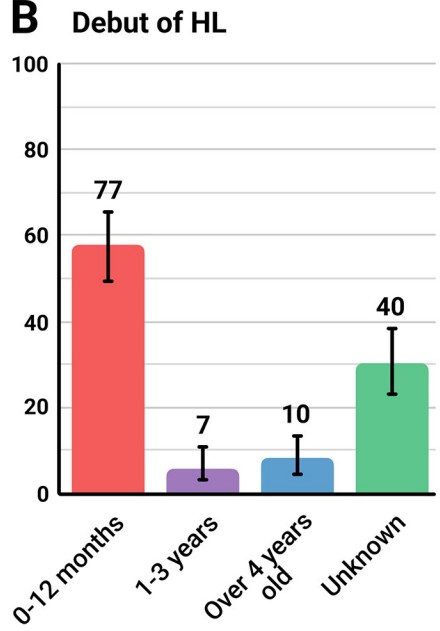

**C  Degree of HL**

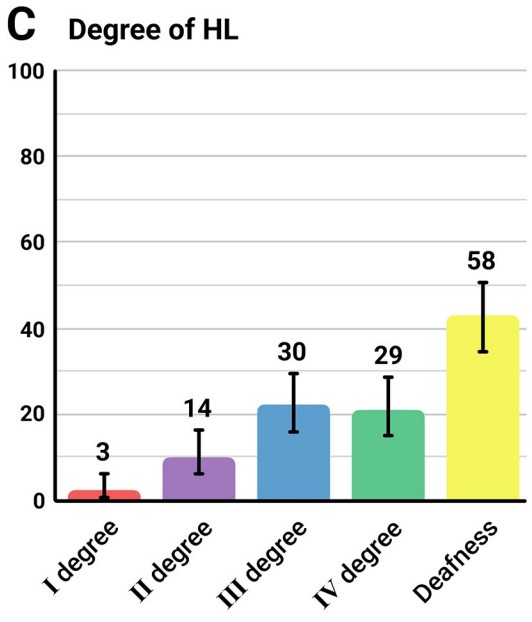

**D  Type of HL**

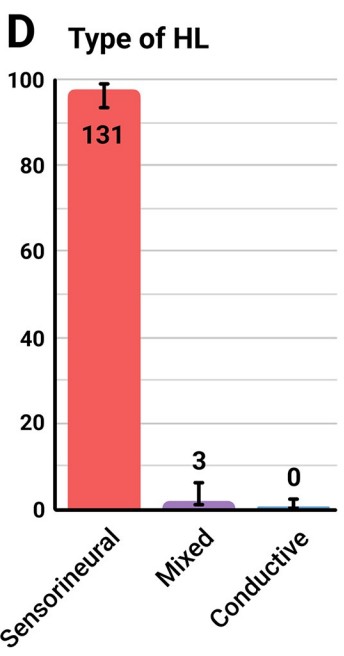

**E  Symmetry of HL**

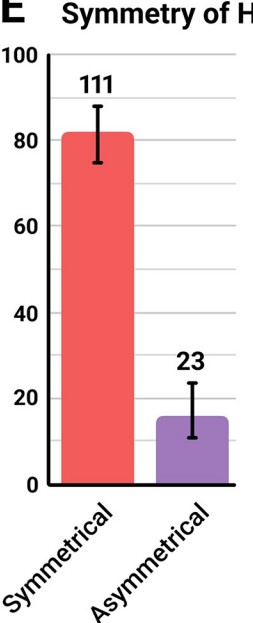

**Fig 3. Clinical and audiological analysis in patients with the c.[-23+1G>A];[-23+1G>A] genotype. Note**: **A**–family history; **B**–debut of HL; **C**–degree of HL; **D**–type of HL; **E**–symmetry of HL.

## Hearing thresholds between male and female patients with c.[-23+1G>A];[-23+1G>A] genotype

We compared hearing thresholds between females (mean age 23.97±15.21 years) and males (mean age 19.72±12.55 years) with the c.[-23+1G>A];[-23+1G>A] genotype. For this comparison, we used the hearing thresholds of both ears. A comparison of the degree of HL in this

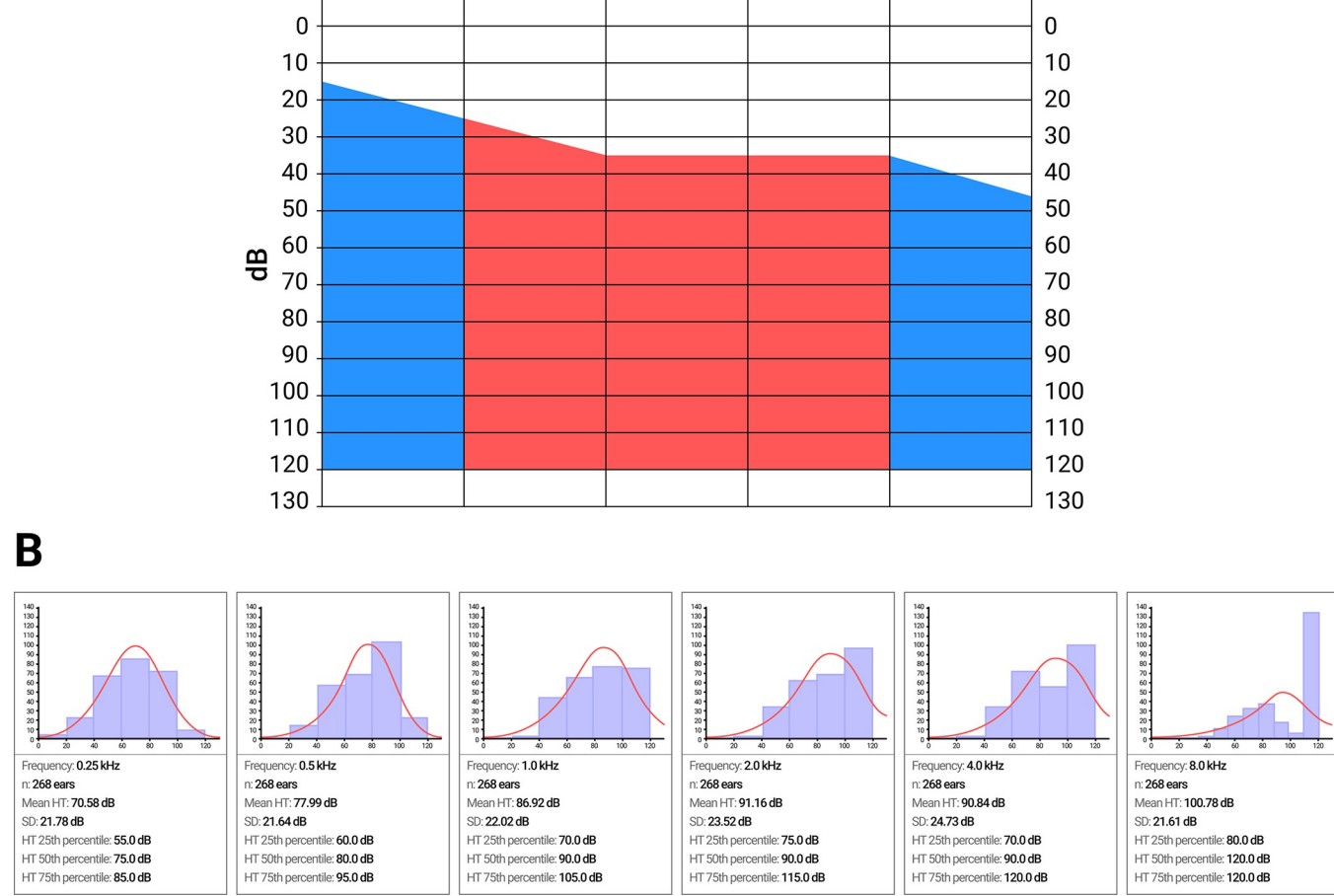

**Fig 4. The audioprofile of the patients with the c.[-23+1G>A];[-23+1G>A] genotype (n = 268 ears). Note: A**—PTA$_{0.5,1.0,2.0,4.0\ kHz}$ is highlighted in red, and the black line is the median hearing threshold. **B**—Audioprofile characteristics at each measured frequency.

sample showed that hearing acuity was better in females than in males. A moderate degree of HL predominated in the female group (48.8% females versus 24.1% in males, $p<0.05$), while profound deafness predominated among males (51.7% males versus 17.1% females, $p<0.01$). The median values of hearing thresholds in PTA$_{0.5,1.0,2.0,4.0\ kHz}$ were 82.34 dB for females and 90.62 dB for males ($p = 0.001$) (Fig 6). In our study, a comparison of the hearing thresholds of males and females at each of the measured frequencies showed that "low" frequencies (0.25 kHz), "speech" frequencies (0.5, 1.0, 2.0, 4.0 kHz) and "high" frequencies (8.0 kHz) are perceived better by females with the c.[-23+1G>A];[-23+1G>A] genotype than males with the same genotype.

## Discussion

### Onset of the disease

Although HL in DFNB1A patients is mostly prelingual, it should not be assumed that the onset is congenital in all patients. This issue is of concern for newborn audiological hearing screening programs because infants who pass the test at birth could develop severe HL within next few months [81]. In our study, among individuals with the c.[-23+1G>A];[-23+1G>A]

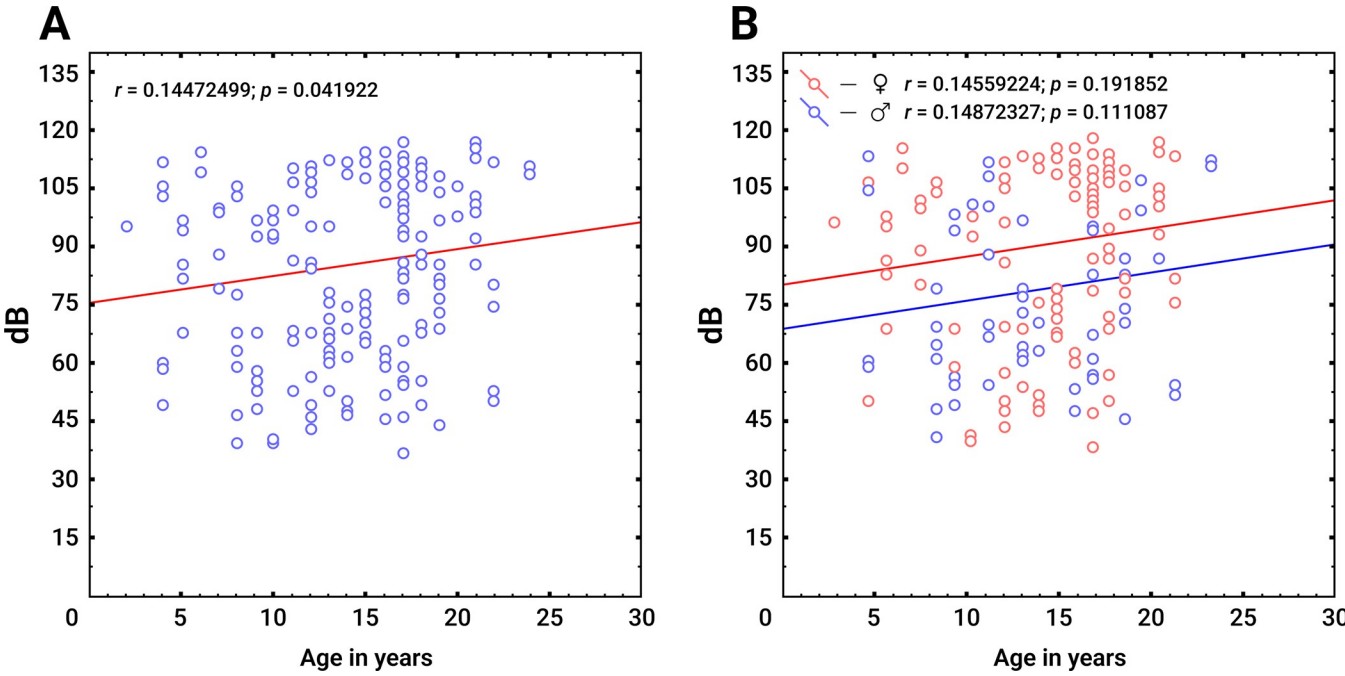

**Fig 5. Regression of hearing thresholds in the PTA$_{0.5,1.0,2.0,4.0\ kHz}$ with age in individuals with the genotype c.[-23+1G>A];[-23+1G>A].** **Note**: **A**–before stratification by sex, n = 198 ears; **B**–after stratification by sex, ♀ –female individuals, n = 82 ears; ♂ –male individuals, n = 116 ears.

genotype, analysis of the age of onset of the disease indicates that in 12.7% (17 out of 134) of patients, HL was diagnosed only after the first year of life (Fig 3B). This finding can be explained by the absence of newborn hearing screening in Russia at the time of birth for the

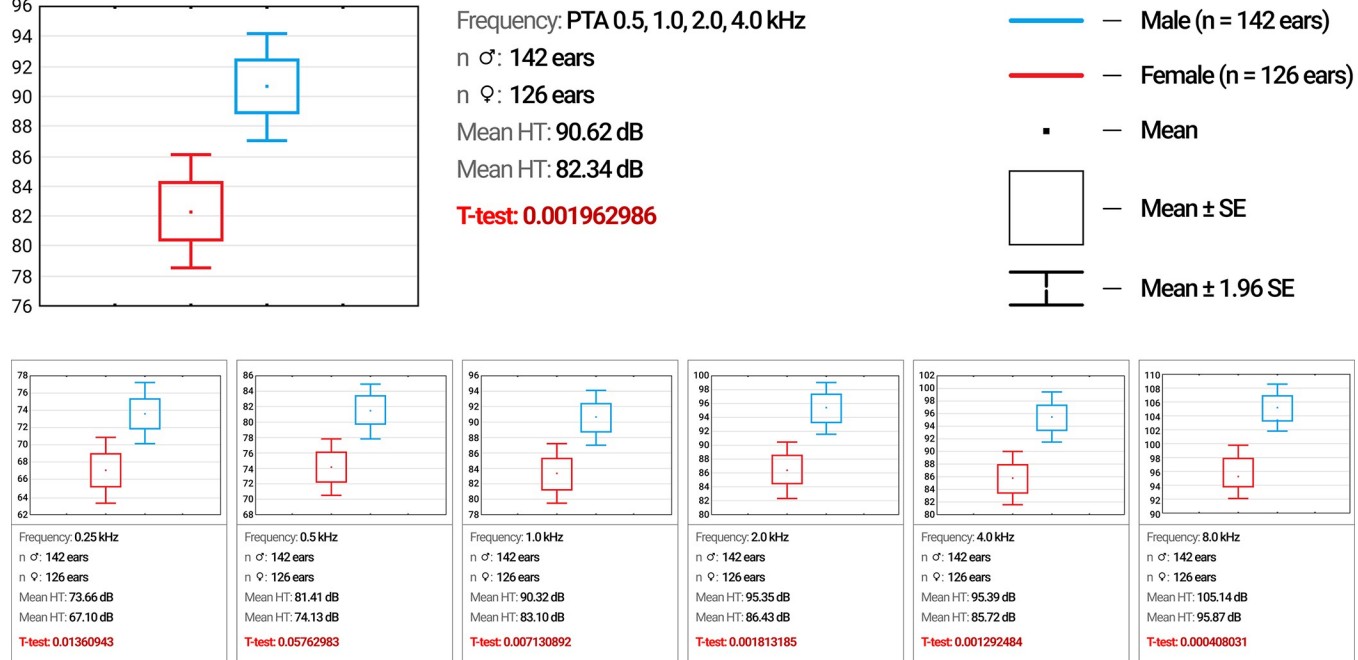

**Fig 6. Hearing thresholds between male and female patients with c.[-23+1G>A];[-23+1G>A] genotype (n = 268 ears).**

majority of examined patients with this allelic variant of DFNB1A (between 1988 and 1998). In addition, we believe that in some cases, a lack of awareness of possible involvement of hereditary factors in deafness among parents [73] could be the reason for the late seeking of medical assistance, since only half of the patients (52.9%) with the c.[-23+1G>A];[-23+1G>A] genotype had a positive family history of HL (Fig 3A). On the other hand, DFNB1A caused by the c.[-23+1G>A] variant of the *GJB2* gene in some cases is not exclusively congenital. The data on DFNB1A patients who passed the newborn hearing screening while being homozygous for other pathogenic *GJB2* genotypes [74, 75] suggests the possibility of later onset of HL in some patients with the c.[-23+1G>A];[-23+1G>A] genotype who also may initially successfully passed the audiological exam.

## Type of HL

Interestingly, the sensorineural type of HL was found in 97.8% of patients who were homozygous for the c.-23+1G>A variant, and the remaining 2.2% had a mixed type of HL (Fig 3D). At the same time, patients with mixed HL had no identifiable otological problems and had normal tympanometry indicators. Based on the clinical features of conductive HL and considering that mixed-type HL was not identified in other *GJB2* genotypes [8, 18, 19], we are inclined to attribute this fact to the patients' mistakes in registering a sound stimulus or vibration of a bone telephone during the audiometry sessions.

## Symmetry of HL

In 82.8% of patients with the c.[-23+1G>A];[-23+1G>A] genotype we detected bilateral symmetrical HL, and in 17.2%, the HL was bilateral asymmetrical (interaural difference in hearing thresholds exceeded 15.0 dB) (Fig 3E). Our findings are consistent with findings reported in a meta-analysis of HL asymmetry in DFNB1A patients with different *GJB2* genotypes, in which the average asymmetry was 14.2% (total range: 0–55.6%) [8]. However, the authors did not explain the causes of this asymmetry since several observed studies did not meet the meta-analysis criteria [8]. Our opinion is that sound perception function is most likely initially symmetrical in all individuals with *GJB2*-related HL. However, we are inclined to believe that the detected asymmetry may be associated with the traumatic effect of excessive amplification by poorly fitted hearing aids, especially if the settings were established in early childhood.

## Audioprofile

One of the most informative audiological characteristics is the audiogram. In most genotype–phenotype studies of DFNB1A, audiograms of patients with different *GJB2* genotypes have variable audiological profiles and are generally divided into three groups: "downsloping", "sloping" and "flat" audioprofiles [8, 18–60]. Previously, "flat" and "sloping" audioprofiles were identified in three British siblings with the c.[-23+1G>A];[-23+1G>A] genotype [76] and among our first reports on 40 patients with this genotype in Eastern Siberia [64]. In the present study, a large cohort of patients with the c.[-23+1G>A];[-23+1G>A] genotype, we confirmed the presence of a very "flat" audioprofile (Fig 4). On average, the slope in $PTA_{0.5,1.0,2.0,4.0 \text{ kHz}}$ was only 5.0 dB (Fig 1). However, in individuals with the c.[35delG];[35delG] genotype, the slope in the $PTA_{0.5,1.0,2.0,4.0 \text{ kHz}}$ was 20 dB (Fig 1). A very "flat" audioprofile in patients with a homozygous c.-23+1G>A splice site variant demonstrated balanced hearing preservation at almost all of the measured frequencies and indicated evenly dispersed damage to the cochlear cells.

## Hearing thresholds in $PTA_{0.5,1.0,2.0,4.0\ kHz}$

The median hearing threshold in $PTA_{0.5,1.0,2.0,4.0\ kHz}$ in patients with the c.[-23+1G>A];[-23+1G>A] genotype was 86.73 dB±21.98 dB (severe HL) (Fig 2). This finding indicates that this splice site variant has a "medium" pathological effect on auditory function. In contrast, the c.35delG variant has more pronounced pathological effect since the median hearing threshold in individuals with the c.[35delG];[35delG] genotype in $PTA_{0.5,1.0,2.0,4.0\ kHz}$ was 103.73 dB ±21.98 dB (profound HL). The c.109G>A p.(Val37Ile) missense variant demonstrated less pronounced pathological effect on auditory function. The median hearing threshold of individuals with the c.[109G>A];[109G>A] genotype in $PTA_{0.5,1.0,2.0,4.0\ kHz}$ was 59.73 dB±21.98 dB (moderate HL). Our results are generally consistent with previous studies, where compound heterozygotes for the splice site variant with nontruncating variants had less severe HL: c.[-23+1G>A];[269T>C] (moderate HL), c.[-23+1G>A];[551G>C] (moderate HL), c.[-23+1G>A];[-254C>T(;)516G>C] (moderate HL), compared to the *GJB2* genotypes where splice site variants in a compound-heterozygous state with truncating variants had a more severe degree of HL: c.[-23+1G>A];[35delG] (profound HL), c.[-23+1G>A];[327_328delGGinsA] (profound HL) [18, 19, 77]. In a multicenter study, the authors [19] noted differences between functional studies [78, 79] and clinical data for the studied splice site variant [19]. Although some authors have suggested that the c.-23+1G>A mutation is associated with mild-to-moderate HL, functional studies in patients with the c.[-23+1G>A];[35delG] genotype did not confirm this assumption because the Cx26 sequence was not detected in the mRNA [78, 79]. Indeed, the pathological effect of splice site variants theoretically must be severe. However, the c.-23+1G>A variant had a "medium" pathological effect on hearing function in our large cohort of patients. In general, this finding reflects our incomplete understanding of the molecular basis for gap junction function in the inner ear and the pathological mechanism of DFNB1A [17]. In the absence of a complete full understanding of the mechanism of DFNB1A, we can only speculate that the very "flat" audioprofile and "medium" pathological effect of this splice site variant on hearing may be due to the presence of normal Cx26 molecules in cochlea (Fig 4). This finding can most likely be explained by the existence of an alternative splicing site in the noncoding region of the *GJB2* gene, allowing the preservation of a certain amount of normal Cx26, even with a disrupted canonical splice site. Currently, only one study on the effect of the c.-23+1G>A variant is known [79]. In that study, RNA was isolated from a lymphoblastoid cell line of one patient with the c.[-23+1G>A];[35delG] genotype [79]. The sequence of this patient yielded only from the c.35delG allele, indicating that the c.-23+1G>A allele was not transcribed or was extremely unstable [79]. To determine the effect of the c.-23+1G>A variant at the RNA level and to test the hypothesis of the existence of alternative splicing sites in the noncoding region of the *GJB2* gene, further extensive studies are needed.

## Progression of HL

We observed a positive correlation of hearing thresholds in $PTA_{0.5,1.0,2.0,4.0\ kHz}$ and age in individuals with the c.[-23+1G>A];[-23+1G>A] genotype (Fig 5). However, HL progression with age may also be due to variable exogenous (noise, ototoxic drugs, trauma, and harmful habits) and endogenous factors (age-related hearing loss and modifier genes) [80, 81]. Currently, many studies have confirmed the progression of HL in patients with various biallelic pathogenic *GJB2* variants [28, 30, 31, 46, 52, 55, 82, 83]. A meta-analysis of 28 studies reporting HL progression data in 1,140 patients revealed that the average progression rate of DFNB1A was 18.7% (range: 0–56.0%) [8]. In a recent study from Shanghai, the incidence of moderate to severe HL among 159 homozygous individuals with the nontruncating "mild" c.109G>A p. (Val37Ile) variant (0.528%, 159/30,122) increased by 9.5%, 23.0%, 59.4% and 80.0% in the age

groups of 7 to 15 years, 20 to 40 years, 40 to 60 years and 60 to 85 years, respectively [56]. Hearing deteriorated by an average of 0.40 dB per year, male individuals were more susceptible, and the deterioration occurred mainly at higher frequencies (4–8 kHz) [56]. The progression HL in individuals with c.109G>A p.(Val37Ile) is also supported by the fact that 43.91% (18/41) of newborns with this *GJB2* genotype successfully underwent hearing screening, which was based on otoacoustic emissions [56]. The results of the correlation analysis obtained in this study indicate that there may be a slow progression of hearing impairment with age in patients with c.-23+1G>A. Overall, we believe that the progression of HL with age may be typical for various pathogenic allelic forms of DFNB1A. However, for patients with severe or profound HL, it is difficult to identify this clinical feature since clinical audiometers are limited by 120 dB, but in patients with mild to moderate HL, it is technically possible.

## Sex differences of HL

In present study, we found sex differences in hearing thresholds for the c.[-23+1G>A];[-23+1G>A] genotype (Fig 6). Previously, there were no reports on sex differences in hearing thresholds in patients with DFNB1A [8, 18–60]. Hearing acuity was reduced in males compared to females at all separately measured frequencies and in the speech frequency range ($PTA_{0.5,1.0,2.0,4.0 \text{ kHz}}$). It is obvious that in patients with the c.[-23+1G>A];[-23+1G>A] genotype, male sex is a risk factor for worsening HL. In our opinion, sex differences in hearing in individuals with the same *GJB2* genotype are mostly associated with cognitive function but not with the degree of cochlear damage in different sexes. It is currently known that the cognitive functions related to verbal, memory and spatial tasks exhibit sex differences [84–88], which are correlated with differences in the volume and proportion of gray matter in the cortex the brain [89, 90]. Analysis of the sex differences in brain gray and white matter in healthy young adults using volumetric segmentation of dual-echo (proton density and T2-weighted) magnetic resonance images confirmed that females temporal lobes in the brain, where the sound signal analyzer is located, at 0.45 mm are thicker [91]. Sex differences in the volume and percentage and asymmetry of the principal cranial tissue may contribute to differences in cognitive functioning related to sound processing in the brain in patients with DFNB1A.

## Limitations of the study

This genotype-phenotype study of the hearing function in patients with biallelic *GJB2* pathogenic variants have a some limitations related with the focusing on the homozygous patients with rare in the world splice site variant c.-23+1G>A, which have a specific audiological features. However, a some common genotype-phenotype findings, such as symmetry, progression and gender differences of the HL may be a typycal for patients with some other *GJB2* variants with mild or moderate pathological effect on hearing function. We hope that this study about hearing function in patients with c.-23+1G>A splice site variant in the *GJB2* gene will be a challenge for other researchers for clearly analyzed these audiological findings in the future studies.

## Conclusions

1. The preservation of hearing thresholds in the speech frequency range ($PTA_{0.5,1.0,2.0,4.0}$ kHz) in patients with the biallelic truncating c.[-23+1G>A];[-23+1G>A] (T/T) genotype is significantly better than in patients with a "severe" truncating c.[35delG];[35delG] (T/T) genotype (*p* = 0.005) and significantly worse than in patients with the nontruncating "mild" c.[109G>A];[109G>A] (NT/NT) genotype (*p* = 0.041), which indicates that this splice site variant has a "medium" pathological effect on hearing function.

2. A detailed clinical and audiological analysis showed that in patients with the c.[-23+-1G>A];[-23+1G>A] genotype, HL is characterized as congenital or early onset (57.5% onset before 12 months), sensorineural (97.8%), bilateral, symmetrical (82.8%), variable in degree of HL (from mild to profound, median hearing threshold in $PTA_{0.5,1.0,2.0,4.0 \text{ kHz}}$ is 86.73±21.98 dB), with an extremely "flat" audioprofile, and with a tendency toward slow progression with age (positive correlation of hearing thresholds with age, $r = 0.144$, $p = 0.041$).

3. In females, the preservation of hearing thresholds in $PTA_{0.5,1.0,2.0,4.0 \text{ kHz}}$ was significantly better (82.34 dB) than that in males (90.62 dB) ($p = 0.001$). Thus, we can conclude that in DFNB1A patients with a homozygous c.-23+1G>A splice site variant, male sex is a factor associated with deteriorating auditory function; in contrary female sex is a protective factor.

## Supporting information

**S1 Fig. Clinical and audiological data of patients with biallelic *GJB2* genotypes.**
(XLSX)

## Author Contributions

**Conceptualization:** Fedor M. Teryutin, Nikolay A. Barashkov.

**Data curation:** Vera G. Pshennikova.

**Formal analysis:** Aisen V. Solovyev, Georgii P. Romanov.

**Funding acquisition:** Sardana A. Fedorova, Nikolay A. Barashkov.

**Investigation:** Vera G. Pshennikova.

**Project administration:** Sardana A. Fedorova.

**Resources:** Vera G. Pshennikova.

**Supervision:** Nikolay A. Barashkov.

**Validation:** Aisen V. Solovyev, Georgii P. Romanov.

**Writing – original draft:** Fedor M. Teryutin, Nikolay A. Barashkov.

**Writing – review & editing:** Sardana A. Fedorova.

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
