## [Decision Letter · Decision Letter 0]

20 May 2024

PONE-D-24-16320Genotype-phenotype analysis of hearing function in patients with DFNB1A caused by the c.-23+1G>A splice site variant in the GJB2 gene (Cx26)PLOS ONE

Dear Dr. Barashkov,

Thank you for submitting your manuscript to PLOS ONE. After careful consideration, we feel that it has merit but does not fully meet PLOS ONE’s publication criteria as it currently stands. Therefore, we invite you to submit a revised version of the manuscript that addresses the points raised during the review process. Please submit your revised manuscript by Jul 04 2024 11:59PM. If you will need more time than this to complete your revisions, please reply to this message or contact the journal office at plosone@plos.org. Please include the following items when submitting your revised manuscript:A rebuttal letter that responds to each point raised by the academic editor and reviewer(s). You should upload this letter as a separate file labeled 'Response to Reviewers'.A marked-up copy of your manuscript that highlights changes made to the original version. You should upload this as a separate file labeled 'Revised Manuscript with Track Changes'.An unmarked version of your revised paper without tracked changes. You should upload this as a separate file labeled 'Manuscript'.

We look forward to receiving your revised manuscript.

Kind regards,

Nejat Mahdieh

Academic Editor

PLOS ONE

Journal Requirements:

"This study was supported by the Ministry of Science and Higher Education of the Russian Federation (FSRG-2023-0003) (AVS., GPR and SAF) and YSC CMP project “Study of the genetic structure and burden of hereditary pathology of the populations of the Republic of Sakha (Yakutia)” (to FMT, VGP, and NAB)."

"The authors declare no competing interests."

Reviewers' comments:

Reviewer's Responses to Questions

**Comments to the Author**

1. Is the manuscript technically sound, and do the data support the conclusions?

Reviewer #1: Yes

Reviewer #2: Partly

2. Has the statistical analysis been performed appropriately and rigorously? 

Reviewer #1: Yes

Reviewer #2: Yes

3. Have the authors made all data underlying the findings in their manuscript fully available?

Reviewer #1: Yes

Reviewer #2: Yes

4. Is the manuscript presented in an intelligible fashion and written in standard English?

Reviewer #1: No

Reviewer #2: No

5. Review Comments to the Author

Reviewer #1: In this study, author present the audiological features of DFNB1A on a large cohort of 134 patients with the homozygous splice site variant c.-23+1G>A , compared with 34 patients with other biallelic GJB2-genotypes. While the topic is interesting and could help to better understanding genotype-phenotype analysis of hearing function in patients with DFNB1A caused by mutations in GJB2 gene, crucial details of how the study was conducted and individuals selected are missing from the methods section.

Reviewer #2: Teryutin et al describe the genotype-phenotype correlation between GJB2 variants and hearing loss presentation with a focus on the -23+1G>A variant which is common to the Yakut population. This study adds to our understanding of the phenotypic variability seen in GJB2 and characterizes the auditory profile for homozygous -23+1G>A individuals. This is an important work for those in the field, however, several items need to be addressed:

1) For most figures, only the -23+1G>A homozygous individuals are described- this visualization should be done for all individuals with the different genotypes to fully appreciate the variability.

2) For the comparison between male and females- do other genotypes exhibit the same skewing in severity? This is critical information to this novel finding

3) I would recommend lumping -23+1G>A/loss of function (frameshift/nonsense variants) compound heterozygous individuals in one category for comparisons as impact to protein is the same.

4) On page 15, the authors state, "This finding is most likely associated with the existence of an

370 alternative splicing site...." Have the authors or other groups attempted to perform RNA assays for this variant from patient cells to corroborate this statement? This would add a critical piece of evidence to this article and tie all the different aspects together.

5) The resolution for the figures should be higher.

6) Major editorial revision for spelling and grammer is stongly recommended.

6. PLOS authors have the option to publish the peer review history of their article (what does this mean?). If published, this will include your full peer review and any attached files.

Reviewer #1: **Yes: **Xue-Dong Song

Reviewer #2: No

---

## [Author Response · Author response to Decision Letter 0]

26 Jun 2024

Author’s response

The authors are sincerely grateful to the reviewers for the thorough review of our manuscript “Genotype-phenotype analysis of hearing function in patients with DFNB1A caused by the c.-23+1G>A splice site variant in the GJB2 gene (Cx26)” (PONE-D-24-16320). We appreciate the dedicated time and effort for providing valuable feedback on our manuscript. All comments have been addressed, with corresponding changes made directly to the manuscript where appropriate. We have highlighted the changes within the manuscript. The revisions have been approved by all authors. Here is a point-by-point response to the reviewers’ comments and concerns.

Reviewer #1: In this study, author present the audiological features of DFNB1A on a large cohort of 134 patients with the homozygous splice site variant c.-23+1G>A , compared with 34 patients with other biallelic GJB2-genotypes. While the topic is interesting and could help to better understanding genotype-phenotype analysis of hearing function in patients with DFNB1A caused by mutations in GJB2 gene, crucial details of how the study was conducted and individuals selected are missing from the methods section.

Author’s response: We added the missing data of the details of the study design in materials and methods section: 

“Data on individuals with HL were obtained from the Republican Hospital # 1 of the National Medical Centre (Yakutsk, Russian Federation), the Republican special residential schools for the deaf and hard-of-hearing children (Yakutsk, Russian Federation) and Deaf individuals included in the Yakutsk Department of the All-Russian Society of the Deaf (Yakutsk, Russia) presented previously [65, 91]. As results the complete sequencing of the non-coding and coding regions of the GJB2 gene was performed in 514 patients with different degree of HL. From 514 patients with HL 168 patients with biallelic GJB2-variants causing of DFNB1A and presenting no objective otological problems were included in this study. DFNB1A patients included in this study were represented by the following GJB2-genotypes: c.[-23+1G>A];[-23+1G>A] (n=134), c.[35delG];[35delG] (n=11), c.[109G>A];[109G>A] (n=2), c.[-23+1G>A];[109G>A] (n=2), c.[-23+1G>A];[35delG] (n=14), c.[35delG];[313_326del14] (n=1), c.[-23+1G>A];[167delT] (n=1), c.[-23+1G>A];[313_326del14] (n=1), c.[-23+1G>A];[333_334delAA] (n=1), and the c.[35delG];[del(GJB6-D13S1830)] (n=1). The largest group in our cohort of patient were ones with the c.[-23+1G>A];[-23+1G>A] genotype (134 patients) named as reference (Ref). For audiological analysis we used the number of ears (168 patients, 336 ears)”.

Reviewer #2: Teryutin et al describe the genotype-phenotype correlation between GJB2 variants and hearing loss presentation with a focus on the -23+1G>A variant which is common to the Yakut population. This study adds to our understanding of the phenotypic variability seen in GJB2 and characterizes the auditory profile for homozygous -23+1G>A individuals. This is an important work for those in the field, however, several items need to be addressed:

Comment 1. For most figures, only the -23+1G>A homozygous individuals are described- this visualization should be done for all individuals with the different genotypes to fully appreciate the variability.

Author’s response: We added the clinical and audiological data about other GJB2-genotypes where it was appropriate. To avoid deviating from the general line focusing on homozygous patients with splice site variant in GJB2 gene we added this data in the supporting Figure S1. 

Comment 2. For the comparison between male and females- do other genotypes exhibit the same skewing in severity? This is critical information to this novel finding.

Author’s response: Unfortunately, we cannot analyze the comparison between males and females patients with other GJB2-genotypes, since our small cohort of patients with other GJB2-variants does not allow this to be done correctly. We are grateful to the reviewer for paid attention to this important moment and we believe that similar gender differences may be is typical for patients with some other GJB2-genotypes. We hope that our study about hearing function in patients with c.-23+1G>A splice site variant in the GJB2 gene will be a challenge for other researchers for analyzed this finding in the future studies.

Comment 3. I would recommend lumping -23+1G>A/loss of function (frameshift/nonsense variants) compound heterozygous individuals in one category for comparisons as impact to protein is the same 

Author’s response: In this work we present the genotype-phenotype correlation of the hearing function focusing on homozygous patients with splice site variant in GJB2 gene. Nowadays this is the most representative sample of patients with rare splice site variant 23+1G>A in the worldwide cohort of patients with hearing loss. Therefore, based on our observations, we believe that the pathological effect on the hearing function of this splice site variant much differed from other truncating variants (TT). Thus, in the presented study we demonstrate that the preservation of hearing thresholds in the speech frequency range (PTA0.5,1.0,2.0,4.0 kHz) in patients with the biallelic truncating c.[-23+1G>A];[-23+1G>A] (T/T) genotype is significantly better than in patients with “severe” truncating c.[35delG];[35delG] (T/T) genotype (p<0.005). In this regard, we think that combining different truncating genotypes (frameshift/nonsense variants) may seriously shift the real phenotypes of hearing function in patients with this splice site variant.

Comment 4. On page 15, the authors state, "This finding is most likely associated with the existence of an

370 alternative splicing site...." Have the authors or other groups attempted to perform RNA assays for this variant from patient cells to corroborate this statement? This would add a critical piece of evidence to this article and tie all the different aspects together.

Author’s response:

Thank you very much for this interesting comment. For our knowledge in this moment known only one work where in order to determine the effect of the c.-23+1G>A variant RNA was isolated from lymphocytes of the one patients with GJB2-genotype - c.[35delG];[-23+1G>A] [Shahin et al., 2002]. We added this moment in discussion section:

“Currently known only one work, where to determine the effect of the c.-23+1G>A variant RNA was isolated from lymphocytes of the one patient with GJB2-genotype - c.[35delG];[-23+1G>A] [78]. The sequence of cDNA from a lymphoblastoid cell line of this patient yielded message only from the c.35delG allele, indicating that the c.-23+1G>A allele was not transcribed or was extremely unstable [78]. To the test the hypothesis about alternative splicing site in the non-coding region of the GJB2 gene further extensive studies are recurred to determine the effect of c.-23+1G>A variant on RNA level”.

Comment 5. The resolution for the figures should be higher.

Author’s response:

The resolution of the pictures is high. Most likely the PDF version does not allow you to convey this. To view the pictures you need to download them in PNG format.

Comment 6. Major editorial revision for spelling and grammer is stongly recommended.

Author’s response:

Fixed.

Reviewer Attachments

In this study, author present the audiological features of DFNB1A on a large cohort of 134 patients with the homozygous splice site variant c.-23+1G>A , compared with 34 patients with other biallelic GJB2-genotypes. While the topic is interesting and could help to better understanding genotype-phenotype analysis of hearing function in patients with DFNB1A caused by mutations in GJB2 gene, crucial details of how the study was conducted and individuals selected are missing from the methods section.

Comment 1. The methods and study details are very vague, there are no details of how the hearing impaired patients were recruited into the study, what was the inclusion/exclusion criteria. How were the final numbers in each cohort established. Was this an observational cohort study or was there specific testing and sample collection a key time points? What was the calendar time for the study?

Author’s response:

Our study is observational cohort study, calendar time of the study (Start: 16.06.2009 End: 24.12.2019). Inclusion criteria are the presents of the biallelic GJB2-variants. Final numbers in each cohort derived by GJB2-genotypes. We added the details of how the hearing impaired patients were recruited into the our study in the M&M section.

 “Sample of HL individuals consisted of patients of the Republican Hospital # 1 of the National Medical Centre (Yakutsk, Russian Federation), students at the Republican special residential schools for the deaf and hard of hearing children (Yakutsk, Russian Federation) and members of Yakutsk Department of the All-Russian Society of the Deaf (Yakutsk, Russia) presented previously [65, 91]. In total, a complete sequencing of the non-coding and coding regions of the GJB2 gene was performed in 514 patients with different degrees of HL. Of them, 168 DFNB1A patients with biallelic GJB2-variants and presenting no objective otological problems were included in this study. Study sample was represented by the following GJB2-genotypes: c.[-23+1G>A];[-23+1G>A] (n=134), c.[35delG];[35delG] (n=11), c.[109G>A];[109G>A] (n=2), c.[-23+1G>A];[109G>A] (n=2), c.[-23+1G>A];[35delG] (n=14), c.[35delG];[313_326del14] (n=1), c.[-23+1G>A];[167delT] (n=1), c.[-23+1G>A];[313_326del14] (n=1), c.[-23+1G>A];[333_334delAA] (n=1), and the c.[35delG];[del(GJB6-D13S1830)] (n=1). The largest group in our cohort of patient were ones with the c.[-23+1G>A];[-23+1G>A] genotype (134 patients) named as reference (Ref). For audiological analysis we used the number of ears (168 patients, 336 ears).”

Comment 2. Line 119, ‘the following GJB2-genotypes’- the level of detail can be reduced as the information is also provided in a table.

Author’s response:

The level of detail was not be reduced because this information is not provided in a table.

Comment 3. Line 196-198, why did the author not choose values of hearing thresholds in normal hearing individuals as a reference

Author’s response:

Since our goal was to compare different GJB2-genotypes in patients with hearing loss, a sample of normal hearing subjects as a reference is not suitable for these purposes.

Comment 4. Line 217, ‘non-truncating (NT/NT anf TT/NT) GJB2-genotypes (n=34)’. Please verify whether there are any writing errors in ‘NT/NT anf TT/NT’.

Author’s response:

Fixed. 

Comment 5. Conclusions. Line 413, authors please describe the conclusions in refined language

Author’s response:

We have refined the language in the conclusions.

Comment 6. English needs to be improved. Many sentences are very wordy and imprecise.

Fixed.

Sincerely yours,

Corresponding author Nikolay A. Barashkov

Head of Laboratory of Molecular Genetics

Yakut Scientific Centre of Complex Medical Problems

barashkov2004@mail.ru

---

## [Decision Letter · Decision Letter 1]

23 Jul 2024

PONE-D-24-16320R1Genotype–phenotype analysis of hearing function in patients with DFNB1A caused by the c.-23+1G>A splice site variant of the GJB2 gene (Cx26)PLOS ONE

Dear Dr. Barashkov,

Thank you for submitting your manuscript to PLOS ONE. After careful consideration, we feel that it has merit but does not fully meet PLOS ONE’s publication criteria as it currently stands. Therefore, we invite you to submit a revised version of the manuscript that addresses the points raised during the review process.

We look forward to receiving your revised manuscript.

Kind regards,

Nejat Mahdieh

Academic Editor

PLOS ONE

Journal Requirements:

Reviewers' comments:

Reviewer's Responses to Questions

**Comments to the Author**

1. If the authors have adequately addressed your comments raised in a previous round of review and you feel that this manuscript is now acceptable for publication, you may indicate that here to bypass the “Comments to the Author” section, enter your conflict of interest statement in the “Confidential to Editor” section, and submit your "Accept" recommendation.

Reviewer #1: All comments have been addressed

2. Is the manuscript technically sound, and do the data support the conclusions?

Reviewer #1: Partly

3. Has the statistical analysis been performed appropriately and rigorously? 

Reviewer #1: I Don't Know

4. Have the authors made all data underlying the findings in their manuscript fully available?

Reviewer #1: Yes

5. Is the manuscript presented in an intelligible fashion and written in standard English?

Reviewer #1: Yes

6. Review Comments to the Author

Reviewer #1: The authors have addressed most of my previous comments. However, there are still a few questions that are not clear, and need to be addressed:

1 Line 197, “PTA0.5,1.0,2.0,4.0 kHz hearing thresholds in patients with 10 different GJB2 genotypes”, the number of cases in some groups in Figure 2 is relatively small. The authors are requested to introduce the statistical method used in this section. Please provide a detailed description of the statistical methods employed in the "Statistical analysis" section at Line 172 or in the "Figure legend" section.

2 In the discussion section, it is recommended that the authors add a discussion on the limitations of this study.

7. PLOS authors have the option to publish the peer review history of their article (what does this mean?). If published, this will include your full peer review and any attached files.

Reviewer #1: **Yes: **Xue-Dong Song

---

## [Author Response · Author response to Decision Letter 1]

10 Aug 2024

Author’s response

The authors are sincerely grateful to the reviewers for the thorough review of our manuscript “Genotype-phenotype analysis of hearing function in patients with DFNB1A caused by the c.-23+1G>A splice site variant in the GJB2 gene (Cx26)” (PONE-D-24-16320R2). We appreciate the dedicated time and effort for providing valuable feedback on our manuscript. All additional comments have been addressed, with corresponding changes made directly to the manuscript where appropriate. We have highlighted the changes within the manuscript. The revisions have been approved by all authors. Here is a point-by-point response to the reviewers’ comments and concerns.

Comments to the Author

Reviewer #1: The authors have addressed most of my previous comments. However, there are still a few questions that are not clear, and need to be addressed:

1 Line 197, “PTA0.5,1.0,2.0,4.0 kHz hearing thresholds in patients with 10 different GJB2 genotypes”, the number of cases in some groups in Figure 2 is relatively small. The authors are requested to introduce the statistical method used in this section. Please provide a detailed description of the statistical methods employed in the "Statistical analysis" section at Line 172 or in the "Figure legend" section.

Author’s response: We added the full description of the statistical analysis in the materials and methods section (Lines 173-187, P.7-8): 

“Statistical analysis of the clinical and audiological data in patients with the c.[-23+1G>A];[-23+1G>A] genotype was performed using the Sampling program, kindly provided by M. Macaulay and adapted by M. Metspalu. Differences of the credible interval at the 95% significance level were considered statistically significant. Comparison of hearing thresholds in PTA0.5,1.0,2.0,4.0 kHz of the reference group c.[-23+1G>A];[-23+1G>A] with other GJB2 genotypes was performed with a Mann–Whitney U test using by software STATISTICA version 8.0 (StatSoft Inc, USA). Differences were considered statistically significant at p<0.05. Correlation analysis of hearing thresholds in PTA0.5,1.0,2.0,4.0 kHz with age in patients with c.[-23+1G>A];[-23+1G>A] genotype was performed with a r-linear regression analysis using by STATISTICA version 8.0 (StatSoft Inc, USA). Differences were considered statistically significant at p<0.05. Statistical analysis of the hearing thresholds between male and female patients with c.[-23+1G>A];[-23+1G>A] genotype was performed with a Student’s t-test using by software STATISTICA version 8.0 (StatSoft Inc, USA). Differences were considered statistically significant at p<0.05”.

2 In the discussion section, it is recommended that the authors add a discussion on the limitations of this study.

Author’s response: We added the limitations of the study (Lines 429-438, P.18).

“This genotype-phenotype study of the hearing function in patients with biallelic GJB2 pathogenic variants have a some limitations related with the focusing on the homozygous patients with rare in the world splice site variant c.-23+1G>A, which have a specific audiological features. However, a some common genotype-phenotype findings, such as symmetry, progression and gender differences of the HL may be a typycal for patients with some other GJB2 variants with mild or moderate pathological effect on hearing function. We hope that this study about hearing function in patients with c.-23+1G>A splice site variant in the GJB2 gene will be a challenge for other researchers for clearly analyzed these audiological findings in the future studies”.

Sincerely yours,

Corresponding author Nikolay A. Barashkov

Head of Laboratory of Molecular Genetics

Yakut Scientific Centre of Complex Medical Problems

barashkov2004@mail.ru

---

## [Editor Report · Decision Letter 2]

13 Aug 2024

Genotype–phenotype analysis of hearing function in patients with DFNB1A caused by the c.-23+1G>A splice site variant of the GJB2 gene (Cx26)

PONE-D-24-16320R2

Dear Dr. Barashkov,

We’re pleased to inform you that your manuscript has been judged scientifically suitable for publication and will be formally accepted for publication once it meets all outstanding technical requirements.

Kind regards,

Nejat Mahdieh

Academic Editor

PLOS ONE
---

## [Editor Report · Acceptance letter]

19 Aug 2024

PONE-D-24-16320R2 

PLOS ONE

Dear Dr. Barashkov, 

I'm pleased to inform you that your manuscript has been deemed suitable for publication in PLOS ONE. Congratulations! Your manuscript is now being handed over to our production team.

Kind regards, 

on behalf of

Dr. Nejat Mahdieh 

Academic Editor

PLOS ONE